# Automatic Modeller of Textile Yarns at Fibre Level

**DOI:** 10.3390/ma15248887

**Published:** 2022-12-13

**Authors:** Desalegn Beshaw Aychilie, Yordan Kyosev, Mulat Alubel Abtew

**Affiliations:** 1Ethiopian Institute of Textile and Fashion Technology (EiTEX), Bahir Dar University, Bahir Dar 1037, Ethiopia; 2Institute of Textile Machines and High Performance Materials, Technical University of Dresden, Hohe Straße 6, Room 141, 01069 Dresden, Germany; 3ENSAIT-GEMTEX Laboratory, Lille University, 2 Allee Louise and Victor Champier BP 30329, CEDEX, 59056 Roubaix, France

**Keywords:** textile fibres, yarn modelling, 3D geometrical model, contact detection, FE simulations

## Abstract

This paper presents a geometrical modelling principle for the modelling of yarns at the fibre level. The woven and the knitted textile structures are built of yarns, which on the other side, are fibrous assemblies. In many yarn and fabric modelling works, yarns are considered as a single line element; however, most yarns are composed of a number of staple or filament fibres. It is then very important to understand the yarn at the micro level for a better understanding, production and application of the above structures. The current paper aims to present the modelling and implementation of yarn structures at the fibre level using the algorithmic geometrical modelling principle. The research work uses basic assumptions for the building of the models and various implementation issues, connected with the proper representation of the single multi-filament yarns, plied yarns and finally the staple fibre yarns. Except for visualization, the generated yarn models are prepared as a basis for mechanical, thermal, fluid flow and other simulations of textile structures using FEM, CFD and other numerical tools.

## 1. Introduction

Yarn, the basic structural element for fabrics is made of a number of textile fibres. Models are used to design, analyse and predict properties of objects and processes. The 3D modelling of yarn reduces the risk, time and cost of designing and enables yarns to be manufactured while satisfying some property requirements. Performing the 3D modelling of a yarn is basic for the construction of fabric models. Even if fibres and yarns are one-dimensional elements, which can be characterized by coordinates of their axis and cross section, only drawing their axis in a 3D space does not produce a realistic enough simulation of products, since single lines have no material-related properties [1]. This work deals with the modelling of a 3D geometrical yarn model at the fibre level.

## 2. State of the Art

The modelling of yarns was started decades ago. The first yarn model was made by Gegauff who derived a simple mathematical relationship between twist angle and yarn strength [2].

Gegauff showed that the extension $\epsilon_{f}$ in an individual filament in a twisted yarn is given by:(1)$$\epsilon_{f}=\epsilon_{y}\cos^{2}\theta$$
where $\epsilon_{y}$ is the extension of yarns as a whole, and θ is the angle of the helix along which the filament lies.

In the second half of the 20th century, the modelling approach was shifted from an empirical approach to the usage of the applied mechanics of fibre assemblies [3].

Platt [4] discussed the variation of the modulus with a twist and gave the following equation:(2)$$\frac{\epsilon_{f}}{\epsilon_{y}}=\cos^{2}a\left[\frac{3\tan^{2}a}{2\left(\sec^{2}a-1\right)}\right]$$

The ratio of moduli was modified by Hearle [5] as:(3)$$\frac{\epsilon_{y}}{\epsilon_{f}}=F_{\left(a,\sigma \right]}$$
(4)$$\frac{\epsilon_{y}}{\epsilon_{f}}=\frac{1}{\left(1+\sigma_{1}\right)}\frac{C^{2}}{\left(1-c^{2}\right)}\left[\frac{4+3\sigma_{1}}{2\left(1+\sigma_{1}\right)}-\frac{1+\sigma_{1}}{2\sigma_{1}}c^{2}-\frac{2\sigma_{1}^{2}+2\sigma_{1^{-1}}}{2\sigma_{1}\left(1+\sigma_{1}\right)}C^{2\left(1+\sigma_{1}\right)}+\log c_{e}\right]$$
where C = cosα, and $\sigma_{1}$ is the Poisson’s ratio of the fibres of the tensile stress.

Three-dimensional textile configurations are modelled based on computer graphics. The 3D modelling of textile composites [6,7,8] and fabrics produced by braiding [9,10], knitting [11,12,13] and weaving [14,15,16,17] at the yarn level have been intensively studied. Most works conducted on 3D yarn modelling [18,19,20] consider a yarn as a single element without considering the fibres from which it is built [21]. Phong or Goround gave a photorealistic effect to the yarn model [22]. Surface technologies have also been applied to model a yarn. Liao and Adanur [23] modelled the 3D geometry of a yarn by dividing the cross section of a uniform cylinder into many arcs by several points. H Lin [24] made a yarn model by combining circles of different diameters. Using a superspline and ellipse approach, Jiang and Chen [25] were able to design arbitrarily shaped yarn cross sections, from whose work TexGen software was developed. Using a cubic B-spline curve, Lin and Newton [26] created a yarn path. Sherburn [27] was also able to model an irregular yarn using a B-splice curve method. To show the twist effect on a yarn model, Zheng [19,28] added a bumping texture on the cross section and Zhao [29] used irregular polygon as a cross-section. To simulate the hairiness effect of a yarn, Zhong [30] added projected triangles.

By rendering the CT scan image of real yarns, cloth rendering models at the fibre level have been studied by [31,32].

Most textile yarns are produced from a combination of fibres. However, in most yarn models as well as fabric modelling works, the yarn is considered as a single cylindrical element applying different changes on its surface and a cross section to show twist and hairiness effects. This work deals with the modelling of a 3D geometrical yarn model at the fibre level.

## 3. Materials and Methods

To model the 3D geometrical model of different yarns, a procedure shown in Figure 1 was followed. Based on the assumptions of a yarn model, yarn geometrical equations were derived, and different modelling parameters were assigned. Using the presented geometrical equations and assigned parameters, Python scripts were developed. Then, the scripts were run in TexGen and TexMind software tools to visualize the models. To analyse and show the application of the models, Ansys workbench was used.

### 3.1. Assumptions and Equations for Yarn Modelling

During yarn formation from fibres, due to twisting, the fibres form a helical structure through the length of the yarn [33,34].

Considering the following assumptions of an idealized yarn proposed by [35] is very important for the modelling of yarns.

–The yarn consists of a large number of fibres of limited length.–The yarn is circular in cross section and regular principally.–The spatial fibre distribution and packing of fibres in the yarn cross section is uniform.–Fibres are assumed to lie on perfect helixes of a constant radius and angle. All those helixes throughout the cross -section have the same number of turns per unit length parallel to the axis of the helix.–The radial location of a given fibre is fixed so that the individual fibres are not migrating between the periphery and interior of the yarn, but stay at a given radial location.–The fibres are assumed to have identical circular dimensions and properties.

At this stage of the process, the different types of yarn production methods, i.e., ring spinning, rotor spinning, etc., were not considered, only how to develop the 3D yarn geometry was. The type of spinning technology is planned for future work. This means that all yarns in one cross section had the same twist per meter.

A yarn consists of large number of fibres in such a way that the fibres at the centre lie straight along the yarn axis and are surrounded by successive concentric cylindrical fibre layers in increasing radii. The fibres in each layer are twisted helically around the preceding layers. The helix angle of twist increases gradually with a radius starting from 0° for the central fibre to α for the surface fibres, and fibres in a given layer have the same helix angle of twist. The twist (turn per unit length) and fibre packing density are constant throughout the length of the yarn [34].

#### 3.1.1. Helical Yarn Model

To define a simple yarn helical model, the following geometrical equations are presented.

Considering the twist given to the yarn, *T* turns per meter indicates *T* number of turns in one meter. The length of the fibre taken for one twist is *h*, and the relation between twist *T* and *h* is expressed as:(5)$$h=\frac{1}{T}$$

When the helical cylinders are opened (Figure 2), flat rectangles are formed. From the rectangular structure,
(6)$$h^{2}+4\pi r^{2}=l^{2}$$
(7)$$h^{2}+4\pi R^{2}=L^{2}$$
(8)$$tan\theta =\frac{2\pi r}{h}$$
(9)$$tan\alpha =\frac{2\pi R}{h}$$
where *r* and *R* are radii of the inner and outer parts of the yarn, respectively.

Concentric circular layers are filled with fibres in contact with each other. The fibres in contact with each other form concentric circular layers in the yarn cross section. The total number of fibres forming the yarn is equal to the sum of the number of fibres in each layer. Therefore, the radius of the yarn *R* is determined by the fibre radius $r_{f}$ and number of the layers *n* that construct the yarn. The thickness of each layer is equal to the diameter of a fibre. From the theory of packing of circles in a circle, the central layer can contain a single fibre or a combination of three fibres (Figure 3) [36,37]. The radius of the yarn at any given layer can be calculated by adding the radius of the first layer with the sum of the thickness of the remaining layers. The radius of the first layer of a model containing a single fibre in the centre is half of the diameter of the single central fibre. For the second case, having three yarns at the centre, the radius of the central layer is $\left(\frac{1+2\sqrt{3}}{3}\right)r_{f}$.

The radius of the yarn in the two cases $R_{1}$ and $R_{2}$ at a given layer is calculated as:(10)$$R_{1}=\left(2n-1\right)r_{f}$$
(11)$$R_{2}=\left(2n+0.154\right)r_{f}$$

As shown in Figure 4, the smaller circles represent neighbouring fibres in a given layer and the radius of the bigger circle represents the radius of the yarn at the given layer. An isosceles triangle is drawn combining the centres of two smaller circles and the larger circle. If the number of fibres in the *n*th layer is *N*,
(12)$$\theta =\frac{360}{N}$$
(13)$$\sin\left(\frac{\theta }{2}\right)=\frac{r_{f}}{R-r_{f}}$$

Substituting $R_{1}$ and $R_{2}$ from Equations (Equation 10) and (Equation 11), the numbers of fibres in the *n*th layers of the two arrangements ($N_{1}$ and $N_{2}$) are:
(14)$$N_{1}=\frac{180}{\sin^{-1}\left[\frac{1}{2\left(n-1\right)}\right]}$$
and
(15)$$N_{2}=\frac{180}{\sin^{-1}\left[\frac{1}{\left(2n-0.84\right)}\right]}$$

In another case, the approximate number of fibres in a given layer can be calculated based on the circumference of an imaginary circle connecting the central points of the fibres in a layer and the radius of the fibres. The circumference of this circle is approximately equal to the summation of the diameters of the fibres in the layer. The radius for this circle is less than the bigger radius by r, ($R=R_{1}-rf)=2r_{f}\left(n-1\right)$.

Dividing the circumference of the circle (2πR) to the diameter of the fibres ($2r_{f}$) and substituting *R* from Equation (Equation 10),
(16)$$N_{1}=2\pi \left(n-1\right)$$

For the other case of having three fibres on the central layer,
(17)$$N_{2}=2\pi \left(n-0.4\right)$$

The fibres in the yarn form helical spiral structure (Figure 5 and Figure 6). For this reason, to model the 3D yarn, the helix parametrization equation is used.

The parametric equations are convenient for describing curves in higher-dimensional spaces. In mathematics, a helix is a curve in three-dimensional space. A three-dimensional curve *u* of a circular helix with radius *r*, helix angle *t* and slope $\frac{b}{r}$ (or pitch = 2πb) is described by the following parametrization [38,39].
(18)$$u\left(t\right)=\left\{\begin{array}{c} x(t)\\ y(t)\\ z(t) \end{array}\right.$$
(19)$$u\left(t\right)=\left\{\begin{array}{c} rcos(t)\\ rsin(t)\\ bt \end{array}\right.$$

The pitch refers to the height *h* of one turn twist. Therefore, the pitch is equal to 2πb. Substituting 2πb in Equation (Equation 5),
(20)$$b=\frac{1}{2\pi T}$$

Since the length in the script is given in unit of mm, the above equation can be rewritten as:(21)$$b=\frac{1000}{2\pi T}$$

Thus, for developing the Python script, the following equation was used.
(22)$$\left\{\begin{array}{c} x(t)=rcos(t)\\ y(t)=rsin(t)\\ z\left(t\right)=\frac{1000}{2\pi T}t \end{array}\right.$$

#### 3.1.2. Geometrical Model for Ply Yarn Axis

After setting the parameters (number of yarns in the ply yarn, fibre diameter, fibre length, twist), we defined a 3D path by a Python script and visualized the model with the TexMind textile viewer [40]. In the Python script, the Python packages (fiberlib, yarnlib and textilelib) developed by Prof. Yordan Kyosev were called.

#### 3.1.3. Geometrical Model for Ply Yarn of Continuous Filaments

Filaments are twisted or plied together to produce a filament yarn. The filament ply yarn is expected to have an equal number of filaments along its length (Figure 7). An equal length of fibres with a certain number can be twisted to produce the filament ply.

In the case of a ply yarn (Figure 8), the helix of the ply axis is generated by the rotation of a vector of length R. The coordinates ($X_{1}Y_{1}Z_{1}$) at a point E can be represented in terms of the angle of rotation θ [41]
(23)$$X_{1}=R\cos\theta ,\;Y_{1}=R\sin\theta ,\;Z_{1}=R\theta \cot\alpha$$

The coordinates of point F, referred to the local axis E where EZ’ is parallel to OZ and EX’ and EY’ are inclined at the angle θ with respect to OX and OY, are
(24)$$X^{\prime }=r_{f}\cos\phi ,\;Y^{\prime }=r_{f}\sin\phi \cos\alpha ,\;Z^{\prime }=-r_{f}\sin\phi \sin\alpha$$

The required transformation to refer these to the fixed coordinate system is
(25)$$\left\{\begin{array}{c} X=X_{1}+X^{\prime }\cos\theta -Y^{\prime }\sin\theta \\ Y=Y_{1}+Y^{\prime }\cos\theta +X^{\prime }\sin\theta \\ Z=Z_{1}+Z^{\prime } \end{array}\right.$$

Substituting Equations (23) and (24) into Equation (Equation 25) yields
(26)$$\left\{\begin{array}{c} X=R\cos\theta +r_{f}\cos\phi \cos\theta -r_{f}\cos\alpha \sin\phi \sin\theta \\ Y=R\sin\theta +r_{f}\cos\phi \sin\theta +r_{f}\cos\alpha \sin\phi \sin\theta \\ Z=R\theta \cot\alpha -r_{f}\sin\alpha \sin\phi \end{array}\right.$$

#### 3.1.4. Geometrical Model for Yarns of Short Staple Fibres

The modelling of staple yarns is more complicated because there are a number of variations along the length of the staple yarn. As shown in Figure 9, the number of fibres on the cross section and the length of fibres at every length step are variable. Therefore, the length and number of fibres should be randomised.

Using the random package of Python, the distribution of the number and length of fibres at different step lengths was randomised. The distribution of the number of fibres can be expressed by a discrete uniform distribution and the distribution of variable lengths of fibres by a continuous random variable distribution [42].

The probability mass function of a discrete uniform random variable number of fibre *X* is given by:(27)$$f\left(X=k\right)=\frac{1}{n},\;k=1,\cdots n$$
N1, N2, N3, etc., indicate the number of fibres at different step lengths (equally spaced checkpoints) on the yarn, and the vertical lines on the figure are imaginary lines to separate the step lengths of a yarn.

The probability density function of a continuous uniform random length of fibre over an interval [*a*, *b*] is given by:(28)$$f\left(x;a,b\right)=\frac{1}{b-a},\;a\leq x\leq b$$

### 3.2. Modelling of Single Yarn Geometry

To model a geometry of yarn from both staple fibres and filaments, based on the assumptions and geometrical equations, a Python script was developed and run by TexGen software.

TexGen models may be used as the basis of simulations for the prediction of a variety of properties, including textile mechanics, permeability and composite mechanical behaviour. TexGen has a Python scripting interface that enables textile models to be reproduced within a mechanics modelling environment in an automatic way [24,43].

The smallest visible structural unit in TexGen is the yarn. In fact, in its interface, it has fibre orientation calculation functions. It is also possible to set fibre parameters for yarn properties including fibre diameter, fibre density, fibre area and number of fibres per yarn for any textile model. However, on a single yarn, no fibres are visible geometrically [24].

In other words, rather than the single element shown in Figure 10, the yarn model was visualized as a combination of a number of fibres which approximated the nature of the real yarn. In this study, the element that was considered as a yarn was reduced to the fibre level by reducing the dimensions (length and diameter) to those at the fibre level. The arrangement of fibres could be visualized from the cross section and surface of the yarn model.

For staple yarn produced from short fibres, by using random length of fibres from 15–30 cm, a two-revolution length of yarn was modelled.

## 4. Result and Discussions

### 4.1. Visualization of the Geometrical Models

After the Python scripts were developed, the visualization of the geometrical models was conducted by two methods.

The first method was using the TexMind viewer. When the scripts were first run on Python software, the output that contained the modelling information was saved as a .CSV file. Then, using the TexMind Viewer, the model was imported as ReCo and visualized. This way, the ply yarn models were viewed by the TexMind viewer (Figure 11 and Figure 12). Figure 11 shows the 3D models of three-ply twisted filament yarns.

In Figure 11b, the filaments were randomly arranged. As result, the arrangement of the filaments differed from the first one (Figure 11a). Staple yarn modelled by randomising the length and number of fibres across the step length was visualized on the TexMind viewer (Figure 12).

The second method used to visualize the yarn models was TexGen software. The Python scripts were developed in Notepad++ and run directly on the TexGen interface. The yarn model containing a number of fibres was viewed in the user interface of TexGen as shown in Figure 13.

The input data for the generation of the models (Figure 13) were the diameter of fibres, number of layers on the model based on the diameter of the yarn, number of twist and degree of angular coordinate. Accordingly, the yarn modelling was performed by setting fibre diameter = 0.01 mm, number of layers = 10, twist = 200 TPM and angular coordinate = 30 × 3.1415/180. The output yarn model had a total of 309 fibres and a diameter of 0.1 mm.

The arrangement of the fibres (Figure 14) shows the effect of the twist at different levels. Each fibre represents the fibres available on their respective layers. For instance, all the fibres in the 7th layer of the yarn have similar arrangement to the 7th fibre and the same is true for the others.

The arrangement of fibres in a single row (Figure 15), containing fibres from every layer indicated how different fibres were positioned on different layer levels.

The fibres were combined together, forming circular layers of different diameters. The first central layer, in one case, had only one fibre circumscribed by six fibres. In another case, the central layer had three fibres circumscribed by nine fibres. The cross-sectional structure of the fibres in each layer of the yarn model is shown in Figure 16.

The figures show how the number of fibres and diameter changed at different layers of the yarn. The first five structures represent the five layers of the yarn model starting with a single yarn at the centre. The others are the 10 layers of a yarn model starting with three fibres at the centre layer.

The total cross section of the yarn model which shows the cross-sectional arrangement of the fibres in the yarn is given in Figure 17.

This figure was taken from the full model yarn shown in Figure 18.

Staple (short-fibre) yarn models for a length of two revolutions run by TexGen are given in Figure 19. The arrangement of fibres with a small amount of twist is given in Figure 19b. Figure 19a shows the random distribution of short fibres during the formation of staple yarns. When the amount of twist was increased, the fibres became more compacted together as shown in Figure 19c.

### 4.2. Application of the Models

The 3D geometrical yarn models at the fibre level can be used for:The visualization of the structure and arrangement of fibres in the yarn. The visualization of the arrangement of filament fibres in filament yarns, single yarns and fibres in ply yarns and the random distribution of short fibres is possible.The simulation of physical properties of the yarn such as tensile, compression and bending properties. Generally, the models can be used for mechanical, thermal, fluid flow and other simulations of the textile structures using FEM, CFD and other numerical tools.The analysis of the contact detection of the fibres in the yarn.Other applications.

To show the application of the models, a contact detection and finite element simulations of the tensile test of the yarn were performed on the models visualized from TexGen software.

The contact detection and finite element simulations of tensile property were conducted in Ansys workbench. For this reason, the TexGen file was exported to an .igs file to be compatible with Ansys. Nylon material was selected, and the boundary conditions were set to be a fixed support on one side and a displacement on the other side.

The contact detection and finite element simulations of the tensile property are shown in Figure 20 and Figure 21, respectively. From the contact detection, it is seen that the fibres in the modelling were not interpenetrated.

The finite element simulation of the tensile test showed that this model could be used for a further analysis of the yarn’s mechanical properties such as bending, compression, etc. The objective of this paper was to model a 3D yarn at the fibre level and to show an application of the models.

A mesh convergence study was conducted by refining the mesh element size without changing the boundary conditions. For the convergence study, six iterations of testing were performed for variable mesh element sizes. The mesh element sizes used were 8 mm, 4 mm, 2 mm, 1 mm, 0.5 mm and 0.25 mm. The test was started with a coarser element size (8 mm) and the other element sizes were half of their preceding element sizes.

The finite element results (Figure 21, Figure 22 and Figure 23) of the tests showed that, except for the first one, the other tests had similar results.

In the first test, the maximum stress was 20.73 MPa, while the maximum stress in the remaining five tests was 50 MPa.

Therefore, the tensile test result converged for mesh element sizes less than or equal to 4 mm.

As shown in the element size versus maximum stress graph (Figure 23), the maximum stresses for mesh element sizes less than or equal to 4 mm did not change.

In similar way, since the models developed are compatible with numerical tools, other mechanical properties of yarns can also be analysed.

## 5. Conclusions

In our current study, for a given diameter of fibres and twist of the yarn, and assuming the yarn and fibres were cylindrical in cross section, the geometrical equations for the given yarns were developed. Later, Python scripts were formulated, and geometrical models were visualized by using TexMind viewer and TexGen modelling software. Randomising the number and length of fibres along the step lengths was developed by using the Python random package. Moreover, the modelling of yarns of short-staple fibres was also possible. TexGen software helped to show the longitudinal structure of every fibre in the yarn during the modelling of the yarn. Moreover, the helical structural change of fibres at different layers and the structure of the yarn at a different level of layers were also observed. This greatly helped to visualize and understand the longitudinal and cross-sectional changes of the yarn at different parameters of yarn formation. In addition, using the model developed, the contact detection and finite element simulations of the tensile behaviour of yarns were performed. A mesh convergence study was performed to analyse how the finite element results were affected by refining the mesh element size.

## Figures and Tables

**Figure 1 materials-15-08887-f001:**
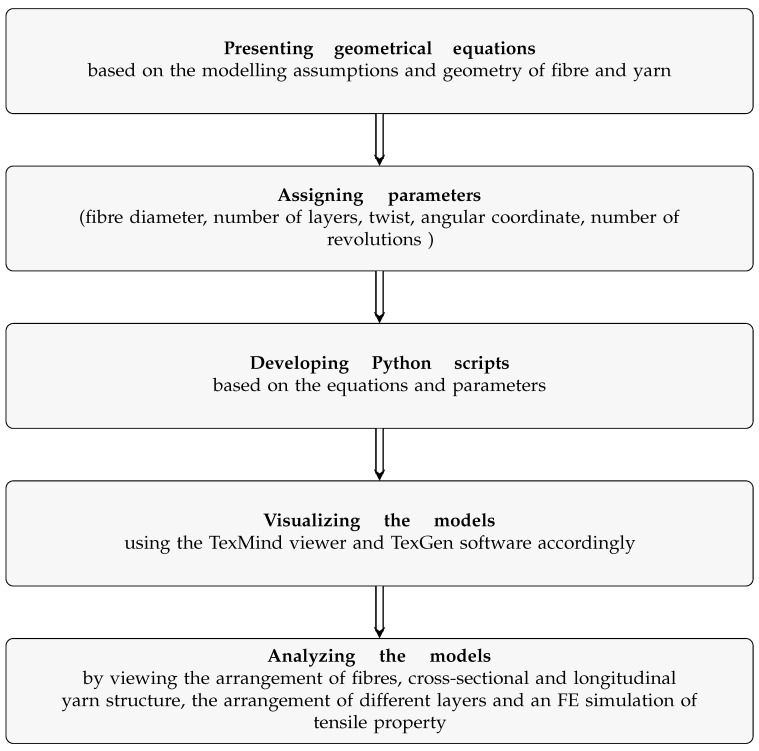
Procedure for 3D geometrical yarn modelling.

**Figure 2 materials-15-08887-f002:**
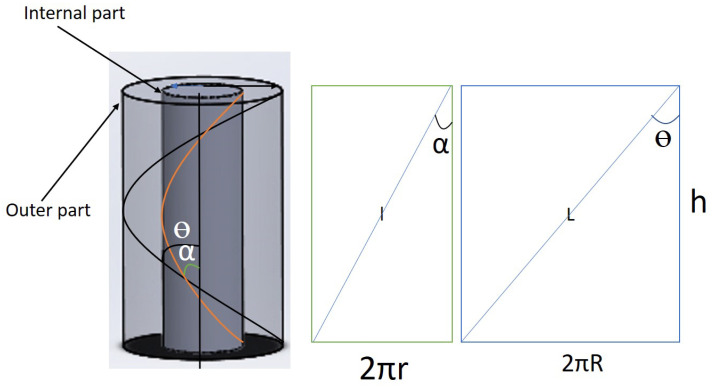
Helical yarn at its different diameters.

**Figure 3 materials-15-08887-f003:**
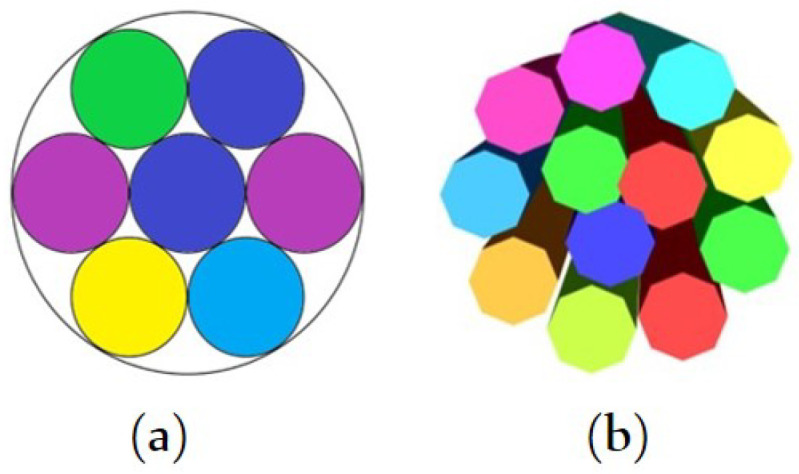
Arrangements with three fibres (**a**) and one fibre (**b**) at the centre.

**Figure 4 materials-15-08887-f004:**
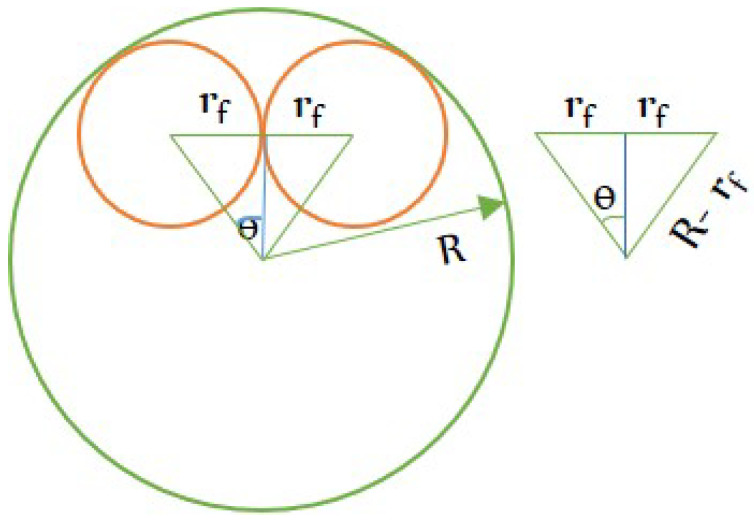
Arrangement of circles in a large circle.

**Figure 5 materials-15-08887-f005:**
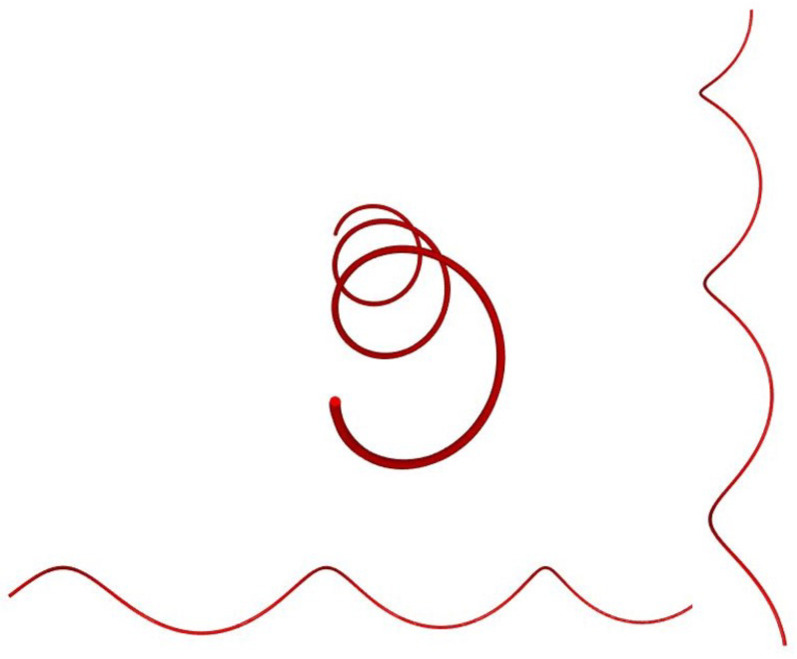
The helical structure of fibres in different views.

**Figure 6 materials-15-08887-f006:**
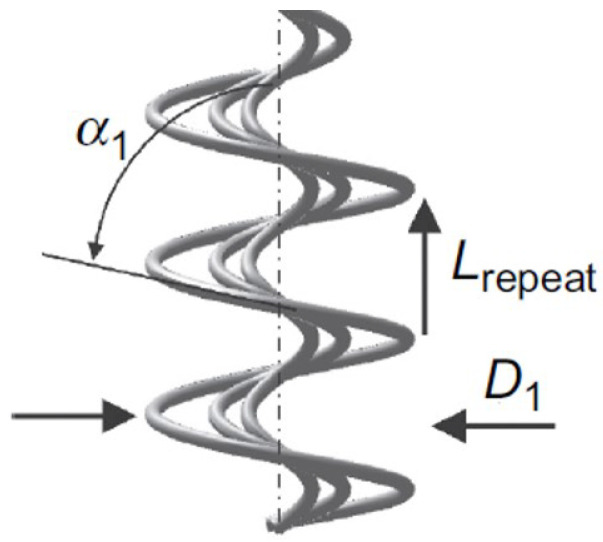
Single yarn paths.

**Figure 7 materials-15-08887-f007:**
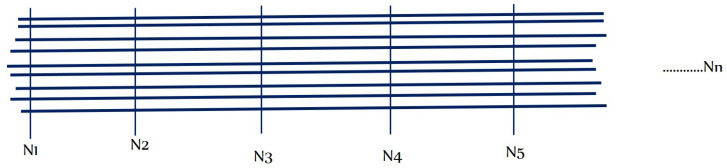
Distribution of filaments.

**Figure 8 materials-15-08887-f008:**
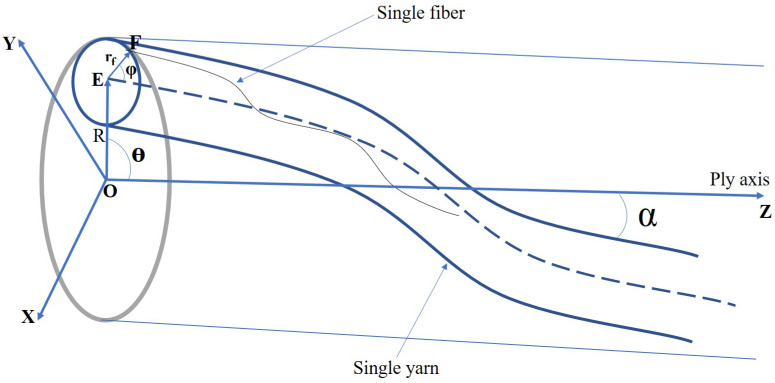
Arrangement of yarns on the plied yarn.

**Figure 9 materials-15-08887-f009:**
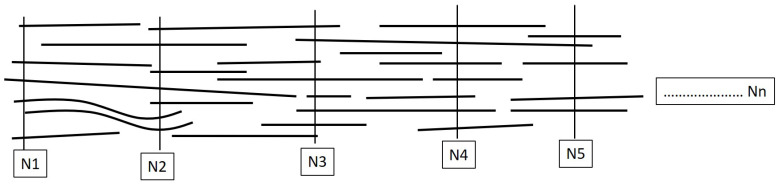
Distribution of staple fibres on yarn.

**Figure 10 materials-15-08887-f010:**
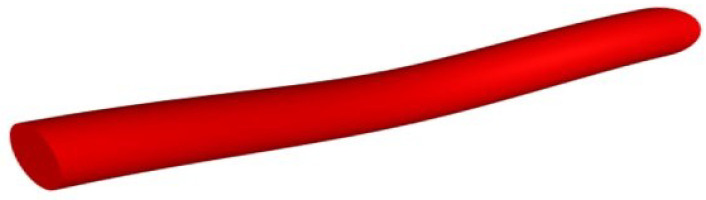
Sample yarn created by specifying 3 nodes with TexGen.

**Figure 11 materials-15-08887-f011:**
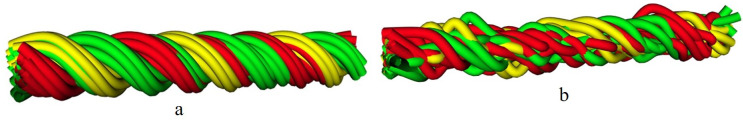
The 3-ply twisted yarn, (**a**) nonrandomised yarn and (**b**) randomised yarn.

**Figure 12 materials-15-08887-f012:**
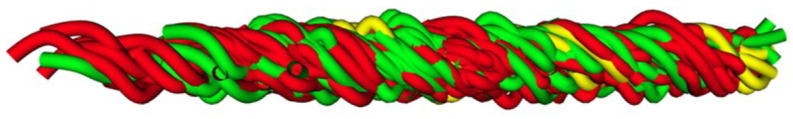
Ply staple yarn model.

**Figure 13 materials-15-08887-f013:**
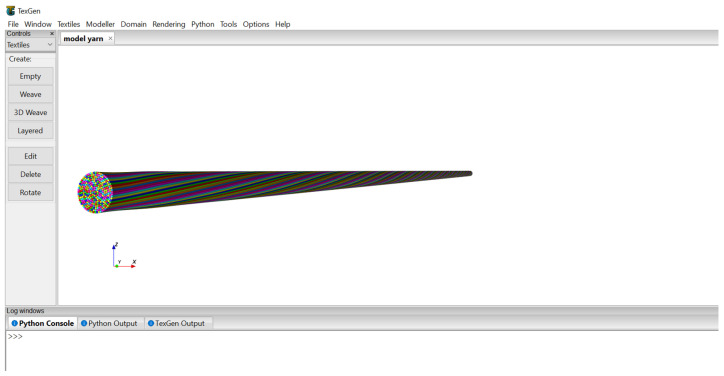
Yarn model in TexGen.

**Figure 14 materials-15-08887-f014:**
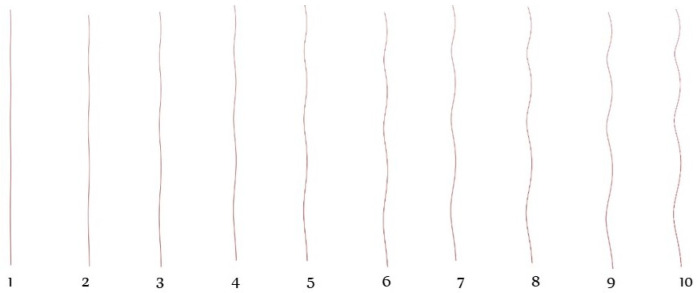
Arrangement of fibres in different layers of the yarn.

**Figure 15 materials-15-08887-f015:**
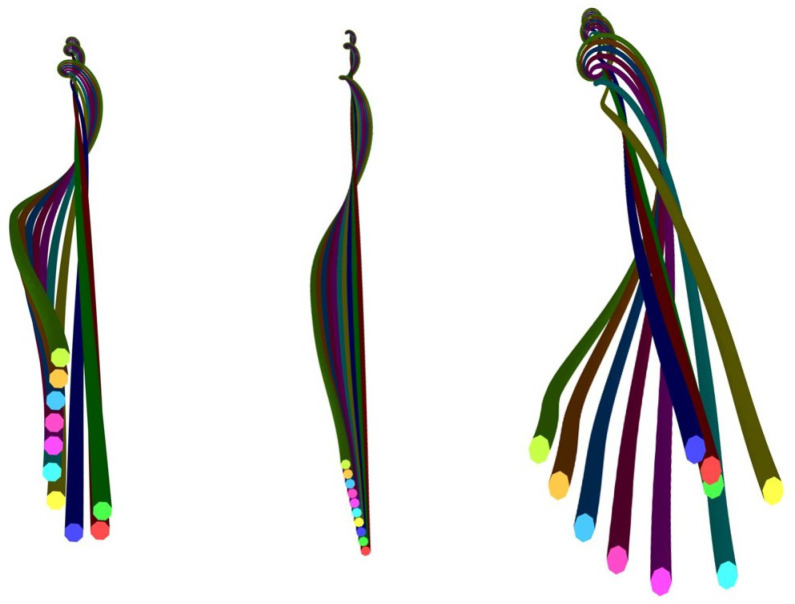
Arrangement of fibres in single row of different layers.

**Figure 16 materials-15-08887-f016:**
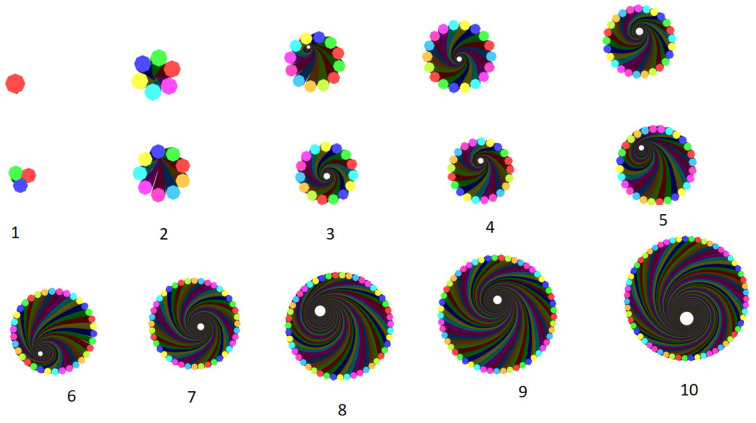
Cross sections of different layers (1–10) of the yarn model.

**Figure 17 materials-15-08887-f017:**
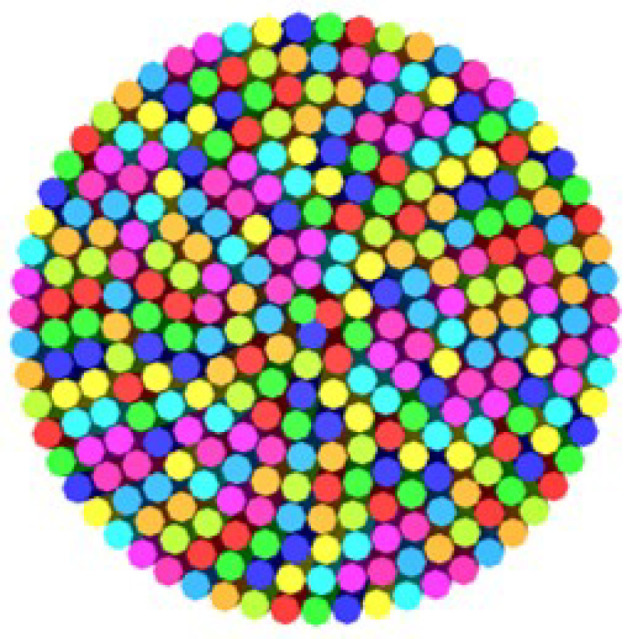
Cross section of the full yarn model.

**Figure 18 materials-15-08887-f018:**
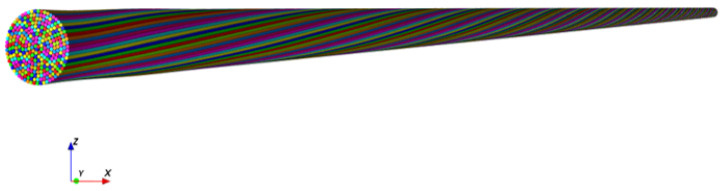
The full yarn model.

**Figure 19 materials-15-08887-f019:**
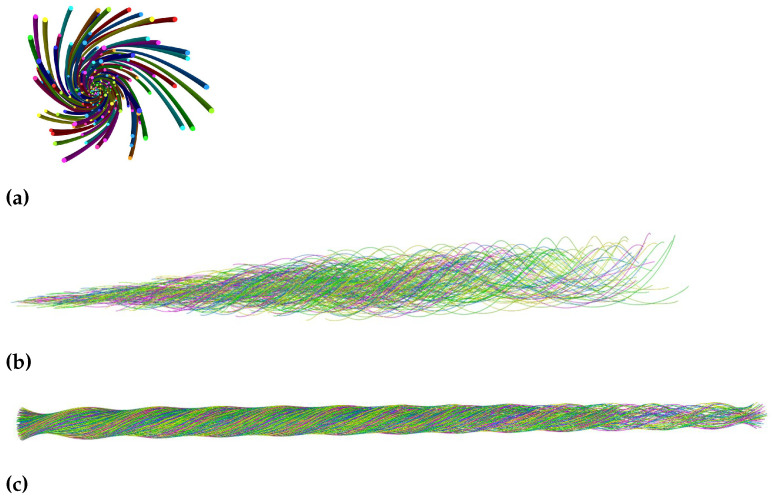
Staple yarn model (Random distribution of short fibre (**a**), staple yarn with small amount of twist (**b**), and with higher amount of twist (**c**)).

**Figure 20 materials-15-08887-f020:**
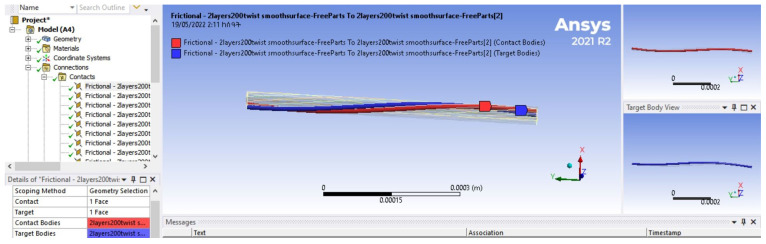
Contact detection.

**Figure 21 materials-15-08887-f021:**
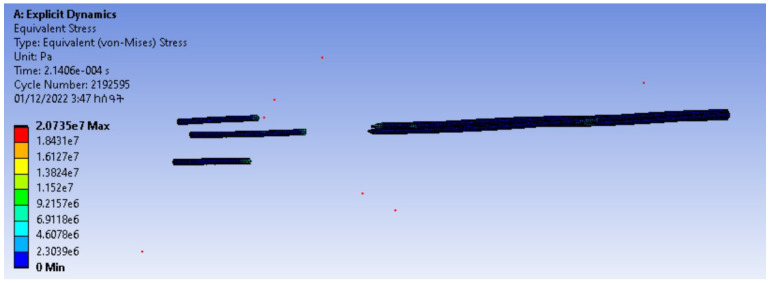
Simulation of the tensile test at a coarser element size.

**Figure 22 materials-15-08887-f022:**
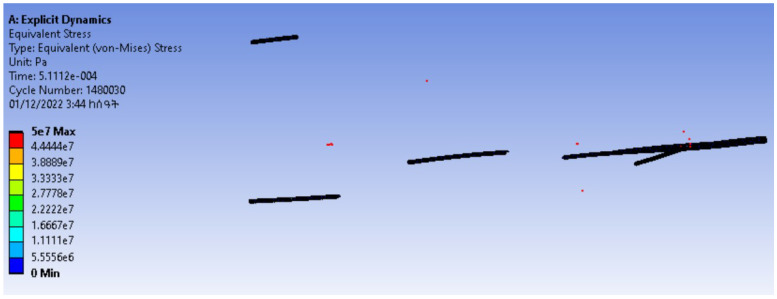
Simulation of the tensile test at element sizes less than or equal to 4 mm.

**Figure 23 materials-15-08887-f023:**
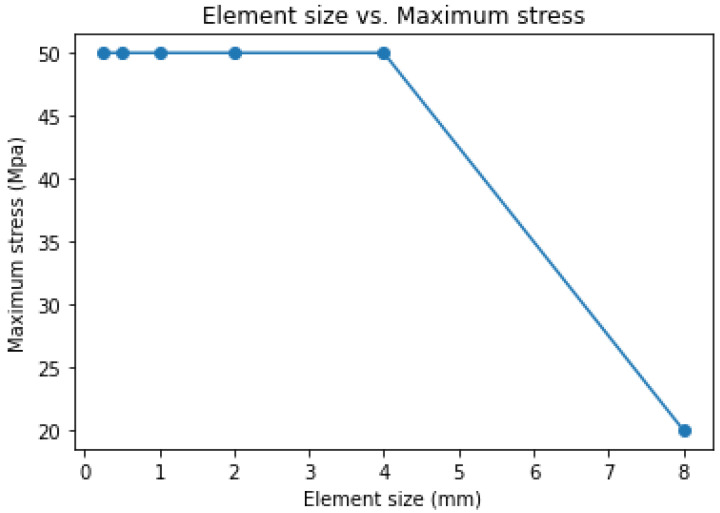
Element size vs. maximum stress.

## Data Availability

Not applicable.

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
