# Peer review of "Automatic Modeller of Textile Yarns at Fibre Level"

_materials, 2022, doi:10.3390/ma15248887_

Round 1

Reviewer 1 Report

Review of the article:

Desalegn Beshaw Aychilie, Yordan Kyosev  and Mulat Alubel ABTEW  „Automatic Modeller of Textile Yarns at Fiber Level” Materials, 2022

Additional comments:

The main goal of the presented research is to increase the accuracy of mapping the 3D structure of textile products. The simulation test results of textile materials presented so far were based on models in which the threads are in the form of monofilament. In fact, each thread making up the fabric is composed of a plurality of fibres, variously arranged to each other. The article presents an attempt to create models in which the threads are composed of many fibres. Therefore, the article is original and fills a gap in the 3D simulation. Although, the authors should provide more details about the scripts developed (or share them) so that readers can use them in their future simulation research. The conclusions are consistent with the evidence and arguments presented
and address the main question posed. The references are appropriate. The following errors were found during the review:

1.      Page 1 row 31 is emperical and should be empirical

2.      Row 274 „2016” should be in bold

Therefore, I suggest that the authors carefully check the content of the article and remove any other errors.

Author Response

Dear respected Editor and Reviewer,

First, we would like to thank and express our great appreciation  for your in-depth and encouraging comments, suggestions, and corrections by reading our manuscript. After completion of the suggested edits, the revised manuscript has benefited from an improvement presentation and clarity. We do hope also this manuscript could benefit the researchers, who work in the field of 3D yarn modelling.

Below, you will find a description of how your comment and suggestions were addressed. In this response document, original reviewer comments and responses are addressed. Moreover, in the revised document, we have made an overall amendment in the whole documents to correct not only the reviewer's comments but also the problems of spelling and grammar. For better understanding and to easily check-up the correction made, Track changes are presented with separate file. 

Comment:

The main goal of the presented research is to increase the accuracy of mapping the 3D structure of textile products. The simulation test results of textile materials presented so far were based on models in which the threads are in the form of monofilament. In fact, each thread making up the fabric is composed of a plurality of fibres, variously arranged to each other. The article presents an attempt to create models in which the threads are composed of many fibres. Therefore, the article is original and fills a gap in the 3D simulation. Although, the authors should provide more details about the scripts developed (or share them) so that readers can use them in their future simulation research. The conclusions are consistent with the evidence and arguments presented
and address the main question posed. The references are appropriate.

Response:

Thank you very much for your understanding of our work objectives. We really appreciate how deep you have read our manuscript and understand the present problem  in 3D geometrical yarn modelling together with our work to address the problem.

Comment:

The following errors were found during the review.

  1.      Page 1 row 31 is emperical and should be empirical

Response:

Thank you very much. It is modified.

Comment:

2.      Row 274 „2016” should be in bold

Response:

Thank you very much! We appreciate your point.
We used the  MDPI Latex template and the bibliography is generated based on the template.  In MDPI reference and citation styles, the year of book, website and proceeding references are not bold. Only the year in journal articles are bold.  (https://mdpi-res.com/data/mdpi_references_guide_v5.pdf)

Kind regards,

Authors

Reviewer 2 Report

The paper is interesting dealing with the modelling of the textile yarns. The methods used serve the parametric development of the models of the yarns. They have an important role for the further mechanical modelling.

Author Response

Dear respected Editor and Reviewer,

First, we would like to thank and express our great appreciation  for your in-depth and encouraging comments, suggestions, and corrections by reading our manuscript. After completion of the suggested edits, the revised manuscript has benefited from an improvement in the overall content, presentation, and clarity. We do hope also this manuscript could benefit the researchers, who work in the field of 3D yarn modelling.

Below, you will find a description of how your comment and suggestions were addressed. In this response document, original reviewer comments and responses are addressed. Moreover, in the revised document, we have made an overall amendment in the whole documents to correct not only the reviewer's comments but also the problems of spelling and  grammar. For better understanding and to easily check-up the correction made, Track changes are presented with separate file. 

Comment:

The paper is interesting dealing with the modelling of the textile yarns. The methods used serve the parametric development of the models of the yarns. They have an important role for the further mechanical modelling.

Response:

Thank you very much for your understanding of our work and encouraging comment.

Kind regards,

Authors

Reviewer 3 Report

Dear Authors,

Authors have introduced the basic mathematical assumptions to construct yarn models and various implementations of fiber level to the proper representation of the single multi-filament yarns, plied yarns and finally the staple fiber yarns. The aim was to prepare a basis to perform several analyses on textile structures, namely mechanical, thermal and fluid flow, using FEM, CFD and other numerical tools. 

The assumptions and implementations performed by the authors in this work have already been published in the literature, namely by the authors of TexGen and TexMind software packages. Therefore, in my opinion this work should be presented as a review paper.

The rate of convergence of the finite element-based method (FEM) concerning Figures (20 and 21) is being remained as an unsolved problem. Spatial convergence for the FEM is defined here as the rate of error reduction with decreasing finite-element size due to the diameters of fiber/yarn and/or deep penetration between yarns, especially in the contact finite-element formulation. Authors must prove convergence of finite-element approximations, such as the ones related to Figures 20 and 21, to qualify the mentioned objectives, implementation, results and conclusion of this work.

In my opinion the authors should consider the possibility to adapt this work and submit it as a review article. In such case, the authors do not need to prove the above-mentioned problem of convergence of the finite element-based method.

Also, in line 61st, the words “Ansys work bench” should be replaced by “Ansys workbench”.

With kind regards.

Reviewer.

Author Response

Dear respected Editor and Reviewer,

First, we would like to thank and express our great appreciation  for your in-depth and encouraging comments, suggestions, and corrections by reading our manuscript. We do hope also this manuscript could benefit the researchers, who work in the field of 3D yarn modelling.

Below, you will find a description of how your comment and suggestions were addressed. In this response document, original reviewer comments and responses are addressed. Moreover, in the revised document, we have made an overall amendment in the whole documents to correct not only the reviewer's comments but also the problems of spelling and grammar. For better understanding and to easily check-up the correction made, Track changes are presented with separate file. 

Comment:

Authors have introduced the basic mathematical assumptions to construct yarn models and various implementations of fiber level to the proper representation of the single multi-filament yarns, plied yarns and finally the staple fiber yarns. The aim was to prepare a basis to perform several analyses on textile structures, namely mechanical, thermal and fluid flow, using FEM, CFD and other numerical tools. 

Response:

Yes, we have presented mathematical assumptions, and developed python scripts; based on the assumptions and mathematical equations presented to model a yarn at the fibre level. The main aim of the work was to model a yarn at the fibre level and show the application of the model in finite element analysis. Preparation of several analyses on thermal, fluid, and mechanical using FEM, CFD and other numerical tools was not the main aim of the work. However, thanks to your comment and new insight, and as we clearly mentioned in section 4.2, we can integrate our model with other tools (CFD) to analyse the different properties like thermal, fluid, mechanical etc.

Comment:

The assumptions and implementations performed by the authors in this work have already been published in the literature, namely by the authors of TexGen and TexMind software packages. Therefore, in my opinion this work should be presented as a review paper.

Response:

Thank you very much for your opinion. The aim is to model a yarn at fibre level. Actually, the papers published by the developers/authors of the TexGen and TexMind software, as software, were to introduce their various applications and the option for the different possibilities of modelling the yarn, or fabrics using the applications. In our paper, basically, the work is done by developing specific python scripts. Of course, we used the TexGen and TexMind software to visualise the developed models as modelling tools. Normally, in TexGen, a single element is considered as yarn to model fabrics. But we minimised the dimensions to the fiber level to generate a yarn composed of fibers in the python scripting. We used TexMind viewer to view the python generated models as ReCo file. The Python scripts developed are our own scripts and the models developed and visualised  which are presented on the manuscript are our own models. 

 In short here are our procedures.

  1. Presenting mathematical equations and assumptions
  2. Developing Python scripts
  3. Running the scripts (some on TexGen and some on Python running software programs like Thonny)
  4. Visualising the models (Some on TexGen and some on TexMind reviewer)
  5. Changing the formats of the models so as to analyse different mechanical properties like FEM.
  6. Performing a tensile finite element simulation to approving of the application of the models.

We believe that such kind of investigation will be categorised as an original paper rather than a review paper. We hope you could see it again and agree with our proposal. 

Comment:

The rate of convergence of the finite element-based method (FEM) concerning Figures (20 and 21) is being remained as an unsolved problem. Spatial convergence for the FEM is defined here as the rate of error reduction with decreasing finite-element size due to the diameters of fiber/yarn and/or deep penetration between yarns, especially in the contact finite-element formulation. Authors must prove convergence of finite-element approximations, such as the ones related to Figures 20 and 21, to qualify the mentioned objectives, implementation, results and conclusion of this work.

Response:

Basically, the objective of this paper is to model a 3D yarn at the fibre level and show the application of the models. Besides, particularly the two figures (Figures 20 and 21) are used to show the application of the models in contact detection and finite element simulation of yarn properties. From these points, we have checked that our models can be used for further simulation of the properties of yarns using FEM, CFD etc. tools. Besides, the finite element simulation of the tensile test shows that this model can be used for further analysis of yarn mechanical properties like bending, compression etc. We understand your concern about resolving the assumption on FM simulation of decreasing finite-element size and working in detail on these areas and focusing on your points in our future work.

Comment:

In my opinion the authors should consider the possibility to adapt this work and submit it as a review article. In such case, the authors do not need to prove the above-mentioned problem of convergence of the finite element-based method.

Response:

Thank you for your opinion. As we discussed above, the finite element is presented in our work to show the application of the models after developing them using Python scripting. We have shown our new yarn models at fibre level with the methodology used and the application of the models. Moreover, our objective is also to model the yarn at the fibre level and show it in the FM simulations. So, in the future, using our models, we can work on solving the problem of convergence as we mentioned above comment.

Comment:

Also, in line 61st, the words “Ansys work bench” should be replaced by “Ansys workbench”

Response:

Thank you very much. It is modified.

Kind regards,

Authors

Round 2

Reviewer 3 Report

Dear Authors,

In relation to the previous comments, the authors have revised the manuscript; however, authors haven’t presented convergence studies based on numerical results for the finite element-based simulation, as introduced in the Figures (20 and 21), for instant, carrying out the comparison between numerical results to the ones in analytical (or experimental) solutions with respect to the configurations these two figures.

Throughout the manuscript, the reviewer also cannot find any constitutive laws, the deformed stress profiles, and/or total strain energies of Figures (20 and 21) to validate the implemented finite-element models and those conclusions such as in case of Section 4.2. It does mean that the statement “Except for visualization, the generated yarn models are prepared as a basis for 10 mechanical, thermal, fluid flow and other simulations of the textile structures using FEM, CFD and 11 other numerical tools” has not yet proven. According to the knowledge of the reviewer, the presented work doesn't carry out scientific contributions emerging in this research topic. The manuscript is not found suitable for publication.

In my opinion the authors should consider the possibility to adapt this work and submit it as a review article. In such case, the authors do not need to prove the above-mentioned problem of convergence of the finite element-based method.

Best regards,

Reviewer

Author Response

Reviewer Comment:

In relation to the previous comments, the authors have revised the manuscript; however, authors haven’t presented convergence studies based on numerical results for the finite element-based simulation, as introduced in the Figures (20 and 21), for instant, carrying out the comparison between numerical results to the ones in analytical (or experimental) solutions with respect to the configurations these two figures.

Throughout the manuscript, the reviewer also cannot find any constitutive laws, the deformed stress profiles, and/or total strain energies of Figures (20 and 21) to validate the implemented finite-element models and those conclusions such as in case of Section 4.2. It does mean that the statement “Except for visualization, the generated yarn models are prepared as a basis for 10 mechanical, thermal, fluid flow and other simulations of the textile structures using FEM, CFD and 11 other numerical tools” has not yet proven. According to the knowledge of the reviewer, the presented work doesn't carry out scientific contributions emerging in this research topic. The manuscript is not found suitable for publication.

In my opinion the authors should consider the possibility to adapt this work and submit it as a review article. In such case, the authors do not need to prove the above-mentioned problem of convergence of the finite element-based method.

Response:

First of all, we would like to thank you for your comments and suggestions.

Even if it was not the initial goal for the current study, based on your constructive comments, convergence test  was  studied for the model in the finite element analysis.

A mesh convergence test was studied by refining the mesh element sizes starting from the courser one. Six iterations of tests were performed varying the elements sizes from 8 mm (the courser) to 0.25 mm (finer).

The results of the tests indicated that except the first test with 8 mm element size, the reaming five tests have similar results.

Therefore it is concluded that for the single type of model analysed, the refining of the mesh element size lower than 4 mm will not change the final result. The more detail work with resulted figures is presented on the manuscript. As far as the model is ready for further numerical analysis using Ansys and other tools, mechanical properties of the yarn can be analysed in similar way.

Regards,

Authors